# Pathophysiological Bases and Clinical Uses of Metalloproteases in Cardiovascular Disease: A Scoping Review

**Laura Manuela Olarte Bermúdez** [1], **Camila Karduss Preciado** [2], **Julián Manuel Espitia Ángel** [2], **Ana María Santos Granados** [3], **Julio Cesar Martínez Lozano** [4], **Carlos Alberto Pacheco Cuentas** [5] **and Diana Marcela Díaz Quijano** [6,*]

1   Fundación Santa Fe de Bogotá, Bogotá 110111, Cundinamarca, Colombia; lauraolbe@unisabana.edu.co
2   Faculty of Medicine, University of La Sabana, Chía 250007, Cundinamarca, Colombia; camilakapr@unisabana.edu.co (C.K.P.); julianesan@unisabana.edu.co (J.M.E.Á.)
3   Spondyloarthropathies Research Group, School of Medicine, University of La Sabana, Chía 250007, Cundinamarca, Colombia; ana.santos@unisabana.edu.co
4   Human Genetics Research Group, School of Medicine, University of La Sabana, Chía 250007, Cundinamarca, Colombia; julio.martinez@unisabana.edu.co
5   Cardiology Group at Clínica Universidad de La Sabana, School of Medicine, University of La Sabana, Chía 250007, Cundinamarca, Colombia; carlospaccu@unisabana.edu.co
6   Health Research Group, School of Medicine, University of La Sabana, Chía 250007, Cundinamarca, Colombia
*   Correspondence: diana.diaz1@unisabana.edu.co; Tel.: +57-311-452-8605

**Abstract:** (1) Objective: Cardiovascular diseases (CVD) are one of the main entities responsible for the progressive increase in morbidity and mortality worldwide. Some of the biomarkers involved in these processes are matrix metalloproteases (MMPs) and disintegrants and metalloproteases (ADAMS), produced by multiple tissues and whose main function is the excessive degradation of the extracellular matrix (ECM). The aim of this study is to describe the existing literature on the role of MMP in the pathophysiology of CVD and its usefulness in clinical practice for the diagnostic and therapeutic approach. (2) Methods: A systematic exploratory review of the literature was carried out according to the guidelines of the Joanna Briggs Institute. The information was collected from the PubMed/Medline and Embase databases, using the search strategy "cardiovascular disease" AND "Metalloprotease". (3) Results: Thirty eight papers that mainly mention 17 types of MMPs were included. Pathologies such as atherosclerosis, coagulation diseases, atrial fibrillation, ischemic heart disease, heart failure, hypertension, dyslipidemias, congenital cyanotic heart disease and Takotsubo cardiomyopathy were identified. (4) Conclusions: The stimulation or inhibition of these biomolecules could generate positive and/or negative effects, which impact the development and prognosis of the disease. Furthermore, they can be potential biomarkers for new diagnostic and even therapeutic approaches in the future.

**Keywords:** metalloproteases; cardiovascular disease; disintegrants and metalloproteases

## 1. Introduction

Cardiovascular disease (CVD) accounts for 40% of deaths from any cause [1]. Cellular aging, genetic factors and unhealthy lifestyles generate functional and structural changes in cardiovascular physiology, related to metabolic diseases and CVD such as the processing of molecules, cytokines, chemokines and growth factors. For this reason, understanding the role they play in different stages of molecular biology is essential to consider them as reliable biomarkers, and together with other prognostic factors, predict their ability to

target therapeutics [2]. Among them, biomarkers of aging and chronic inflammatory state such as matrix MMPs, which can be used in references to cardiac and vascular tissue [3,4].

MMPs are proteolytic enzymes that are responsible for the degradation of proteins in the ECM and regulate both their accumulation and decrease in tissues. Excessive degradation of the ECM is regulated by tissue metalloproteinase inhibitors (TIMPs). In addition, these enzymes have been established together with producing cells such as endothelial cells, fibroblasts and leukocytes that will contribute to maintaining a regulated activity of MMPs [3,5].

The imbalance between MMPs and TIMPs produces dynamic changes in cardiovascular physiology by various extracellular and intracellular mechanisms, for example, an increase in signaling and activation of vascular smooth muscle cells responsible for generating structural and functional support will lead to an accumulation of ECM and excessive vascular remodeling [5,6]. The remodeling of blood vessels related to CVD involves the generation of a pro-inflammatory profile in the ECM and an increase in some mediators such as angiotensin II (AngII), endothelin, mineralocorticoid receptor signaling molecules, transforming growth factor B1 (TGFB1), tumor necrosis factor (TNF) alpha and MMPs [3,5,7]. Through a scoping review, this study aimed to identify and present current information about the relationship of MMPs in the pathophysiology and development of CVD, as well as their usefulness in clinical practice for the diagnostic and therapeutic approach.

## 2. Materials and Methods

A scoping review was conducted in accordance with the rigorous framework outlined by the Joanna Briggs Institute (JBI) and the PRISMA-ScR statement (Preferred Reporting Items for Systematic reviews and Meta-Analyses extension for Scoping Reviews) [8,9]. The review process followed these systematic steps: (a) clearly defining the objectives of the review; (b) developing a detailed and reproducible protocol; (c) conducting a comprehensive search to identify relevant studies across multiple databases; (d) screening and selecting eligible studies, followed by data extraction based on predefined criteria; and (e) synthesizing and presenting the findings in a structured format. The study protocol was registered internally at the University of La Sabana with the code MED-346-2023.

The review aimed to address the following research question:

1. What is the existing scientific evidence on the role of MMPs in the pathophysiology of CVD, particularly in their clinical uses and their potential as future therapeutic targets?

### 2.1. Eligibility Criteria

The inclusion criteria were: (a) all publications with a central focus on CVD and MMPs in humans, (b) studies in English and Spanish, (c) studies that discuss the pathophysiology or clinical use. The exclusion criteria were: (a) studies in animals.

### 2.2. Selection of Sources of Evidence

The search was carried out on 24 June 2023, we included the PubMed/MEDLINE and Embase databases using Boolean operators and keywords according to the data system. The following search strategy was used for Pubmed and adapted to the other databases: "Cardiovascular disease" AND "Metalloprotease" filter: 2023.

The search and selection of studies was carried out by the researchers blindly and independently. Initially, duplicate records due to overlap between the consulted databases were detected in the Rayyan® web tool and were suppressed. Subsequently, the same tool was used to screen by title and abstract. Discrepancies in initial screening were compared and resolved by a consensus made by the investigators. Relevant identified articles were

retrieved in full text for in-depth reading by the researchers independently to define their final entry. The discrepancies were resolved by consensus.

*2.3. Synthesis of the Results*

The following information was extracted from the articles included in this study: title of the article, author, year, country, type of study, and clinical and physiopathological characteristics. For the synthesis and analysis of the information, the data were organized and categorized into type of CVD, type of MMP, physiopathology and therapeutic approach. The information was synthesized according to the objective of the studies and the characteristics of the population.

The results of the review were organized according to the categories established by Grudniewicz et al. [9]. Two tables were elaborated to provide an overview of the studies, followed by a narrative synthesis of the most relevant findings (Tables S1 and S2).

## 3. Results

One hundred and seventy six articles were identified. n = 41 were eliminated due to duplicates. Subsequently, n = 84 were excluded from initial screening mainly due to incorrect population and other outcomes. After an in-depth reading of the selected articles, n = 12 articles were excluded at this time, because they were animal studies (n = 8), they addressed pathologies other than cardiovascular disease (n = 2), they were not based on the pathophysiology and therapeutic target of ADAMS and/or MMPs (n = 1), nor did they mention any of these molecules throughout the study (n = 1). Including n = 38 complete documents in the final analysis (Figure 1). This section may be divided by subheadings. It should provide a concise and precise description of the experimental results, their interpretation, as well as the experimental conclusions that can be drawn.

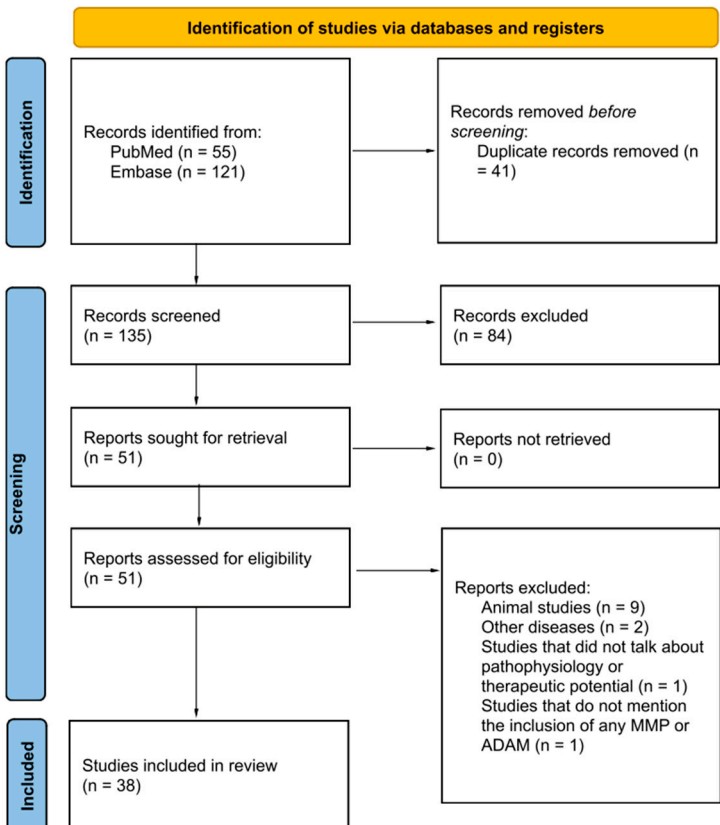

**Figure 1.** PRISMA flow diagram.

Considering the type of study, 20 of the articles reviewed were narrative reviews, nine articles were studies in which in vitro experiments were performed, five additional articles were observational (analytical) longitudinal prospective cohort studies, two were cross-sectional studies, one was a randomized clinical trial and finally there was one "before and after" study. Additionally, no systematic literature reviews were found on this topic.

The studies reviewed were conducted between 1996 and 2025. Seven articles correspond to the period between 1996 and 2010, and 31 articles to publications from 2011 to 2025.

The most frequent cardiovascular pathology was atherosclerosis. Likewise, the most frequent coagulation diseases were Thrombotic Thrombocytopenic Purpura (TTP) and Von Willebrand Disease. Other diseases included acute coronary syndrome (ACS—ischemic heart disease, cardiomyopathies, atrial fibrillation, stroke, heart failure, high blood pressure, dyslipidemia, aneurysms and aortic dissection.

In this study, 17 molecules were grouped into MMPs with six subtypes (1, 2, 8, 9, 10 and 4/17), ADAM(TS) with nine subtypes (1, 2, 3, 6, 7, 10, 12, 13 and 17) neprilysin and CD10. Finally, regarding MMPs as a therapeutic target in CVD, 21 articles mentioned the possible therapeutic potential of these molecules, by their stimulation and/or inhibition (Table S1).

Table 1 provides a summary of the main findings of MMPs and ADAM(TS) according to their role in pathophysiology and the mechanism by which these molecules work as potentials therapeutics in cardiovascular disease.

**Table 1.** Role of the different agents in pathophysiology and therapeutic targets regarding CVD.

| Molecule's Name | Pathophysiology in CVD | Therapeutic Target |
|---|---|---|
| ADAMTSL 1 | Associated with cardiac fibrosis and ECM remodeling. | |
| ADAMTSL 2 | Regulates TGFB, reduces its activation and it is linked to dysplasias. | |
| ADAMTSL 3 | Cardioprotective, reduces TGFB and collagen. | TGFB inhibitor, fibrosis biomarker. |
| ADAMTSL 6 | Inhibits TGFB, affects aortic dilation and blood pressure. | |
| ADAMTSL 7 | Pro-atherogenic, degradates COMP. | Therapeutic inhibition via miRNA or blockers. |
| ADAMTSL 10 | Regulated by calcitriol, activates sRAGE, involved in inflammation. | Inhibition reduces inflammation, calcitriol-based therapies. |
| ADAMTSL 12 | Associated with cardiac hypertrophy, vascular remodeling. | |
| ADAMTSL 13 | Regulates VWF, low levels = ↑ cardiovascular mortality. | Recombinant rADAMTS13 in TTP, biomarker for MI and stroke. |
| ADAMTSL 17 | Promotes inflammation and remodeling, activated by catecholamines. | Inhibition reduces inflammation, modulated by Rhom1-2. |
| MMP 1 | Degradates collagen I and LDLr, linked to plaque rupture. | Inhibition enhances LDL uptake, possible CV prevention. |
| MMP 2 | Induced by IL-17, involved in angiogenesis and adipogenesis. | Nanomaterials reduce its expression (anti-inflammatory). |
| MMP 8/9 | Associated with RANKL, coronary calcification. | Inhibition via ERK1/2 and PI3K/Akt pathways. |
| MMP 10 | Vascular remodeling, unstable plaques, inflammation. | Inhibition may prevent atherosclerotic progression. |
| MMP 17/MT4-MPP | Remodeling, inflammation, aneurysms, cell migration. | Potential therapeutic target in atherosclerosis. |
| Neprilysin | Degrades natriuretic peptides, promotes vasoconstriction. | Inhibition has antihypertensive effect. |
| CD10 | Degrades both vasodilators and vasoconstrictors. | Inhibition enhances natriuretic peptide effects, useful in heart failure. |

↑ Increase.

### 3.1. Pathophysiology of ADAMTS and MMPs in Cardiovascular Disease

#### 3.1.1. ADAMTSL 1

ADAMTSL 1 locus is linked to cardiac fibrosis in diseases such as diabetes mellitus, renal disease, aortic stenosis, cardiomyopathy, heart failure, atrial fibrillation, and rheumatoid arthritis. It is expressed especially in fibroblast. Some variants are enriched in skeletal muscles, aorta and intracranial aneurysms suggesting an important role in cardiovascular tissue; however, it is downregulated in intraabdominal aortic aneurysms [10]. This molecule is also related to cardiac ECM remodeling.

#### 3.1.2. ADAMTSL 2

Transforming growth factor beta (TGFB) cardiac activity is regulated by ADAMTSL 2. Its expression is strongest in the heart followed by the lungs, nerves, arteries, liver, kidneys and other endocrine organs. Indeed, half of Geleophysic dysplasia cases are caused by ADAMTSL2 mutations. Consequently, this mutation also gives rise to the more severe Al-Gazali skeletal dysplasia, with cardiovascular defects of pulmonary artery hypoplasia

and stenosis, aortic stenosis and perimembranous VSD. The molecular mechanism supports a role for ADAMTSL2 in suppression of TGFB activity, as ADAMTSL2 localizes to microfibrils and binds directly to fibrillin-1 and fibrillin-2 in fibroblasts. The overexpression of ADAMTSL2 in cardiac fibroblasts reduces TGFB activation. However, ADAMTSL2 seems to be part of the sophisticated TGFB feedback system [10] as TGFB upregulates ADAMTSL2 expression.

### 3.1.3. ADAMTSL 3

ADAMTSL 3 shows a high expression in heart and vasculature, being cardioprotective in the failing heart, by the overexpression in cardiac fibroblasts resulting in reduced TGFB activity and myofibroblast differentiation, as well as reduced collagen deposition. Some ADAMTSL3 variants have growth-limiting functions; some of them are enriched in patients with intracranial aneurysms, and some others reduced in both intracranial and abdominal aortic aneurysms [10].

### 3.1.4. ADAMTSL 6

These proteins were detected in ECM of arteries, skin, tendons, cartilage, and the mitral valves of the heart. Mechanistically, ADAMTSL6β binds to the N-terminus of fibrillin-1, and both isoforms promote fibrillin-1 fibrillogenesis. ADAMTSL6β also inhibits active TGFB. Recent studies have linked ADAMTSL6 variants to aortic dilation, aneurysms, aortic size, blood pressure and spontaneous coronary artery dissection. Reduced ADAMTSL6 expression correlates with increased TGFB activity and fibroblast activation in tumor tissue [10], emphasizing the role of ADAMTSL6 in regulating TGFB.

### 3.1.5. ADAMTS 7

ADAMTS 7 was strongly associated with CAD (coronary artery disease) and promoted atherosclerosis [11]. ADAMTS7 accelerated the progression of cardiovascular disease by promoting the breakdown of the oligomeric matrix of cartilage (COMP) [12]. The gene was found in samples from human adults with the transcription 5.5 kb in the heart, pancreas, kidneys, skeletal muscle and liver. Individuals carrying a single nucleotide polymorphism (SNP) in the ADAMTS 7 gene had a 19% increased risk of developing atherosclerotic disease (involved in the development and early progression of atherosclerosis but not in thrombosis) [13].

### 3.1.6. ADAMS 10

Calcitriol stimulated the activity of the ADAMS 10 promoter in human blood cells and induced the translocation of ADAMTS10 to the plasma membrane, that triggered the elimination of TNF receptor 1 ectodomain in smooth muscle vessels, with increased sRAGE production [14]. On the other hand, inhibition of ADAMS 10 increased AngII-mediated CyclinD promoter induction [15], likewise, ADAMS 10 increases its expression during atherosclerotic plaque progression by macrophages and foam cells [16].

### 3.1.7. ADAMS 12

Increased ADAMS 12 expression levels were found in the heart tissue of patients with obstructive hypertrophic cardiomyopathy and in mice with AngII-induced hypertension and cardiac hypertrophy. It is possible that a similar chain of events lead to SnoN degradation during fibrosis associated with cardiac hypertrophy, which would explain the high levels of ADAMS 12 expression in hypertrophic myocardium [17]. It exposes ANG II and its AT1 receptor inducing vascular remodeling: hyperplasia, hypertrophy, and vascular smooth muscle cell migration [18]. Inhibition of ADAMS 12 reduced the induction of the AngII-mediated atrial natriuretic peptide promoter, as well as attenuated

the AngII-mediated induction of myosin-2v light chain promoter activity. ADAMS 12 may also play a role in inducing α-skeletal actin expression [15].

### 3.1.8. ADAMTS 13

Metalloprotease and disintegrant ADAMTS 13 cleaved large von Willebrand factor (VWF) multimers into smaller and less coagulant forms. In individuals with low ADAMTS 13 activity, the risk of cardiovascular mortality was higher than that of people with high ADAMTS 13 activity [19–21]. The effects of functional ADAMTS 13 deficiency were associated with several pathologies including thrombotic thrombocytopenic purpura, cardiovascular disease, and inflammation [11,22]. Patients with severe ADAMTS 13 deficiency showed lower platelet counts, less kidney dysfunction, greater presence of neurological abnormalities and hemolysis [23], as well as high residual platelet reactivity (RPR) [24].

In patients with an ADAMTS 13 level below 38%, the presence of ultra-large VWF were high. Also, the relative risk of ischemic heart disease was 18.2%, and the relative risk of stroke was 11.5%. Measuring VWF and ADAMTS 13 could be a good indicator of the likelihood of acute myocardial infarction (AMI) and stroke [25]. In the same way, some alleles of the ADAMTS 13 gene were associated with negative cardiovascular effects and a higher frequency of developing CVD [26].

In children with cyanotic congenital heart disease, low preoperative ADAMTS 13 activity, activated partial thromboplastin time, and the need for cardiopulmonary bypass was correlated with postoperative bleeding. In the postoperative period, ADAMTS 13 activity increased. Changes in circulating ADAMTS 13 suggested enzyme consumption, associated with abnormal VWF structure and function [27].

### 3.1.9. ADAMS 17

ADAMS 17 can cleave several cell surface molecules, including interleukin (IL) receptor 6, vascular cell adhesion molecule 1, and L-selectin, facilitating the infiltration of leukocytes through the vasculature into adipose tissue, promoting metabolic inflammation, presumably through the release of inflammatory cytokines such as TNF, and regulating pathways involved in the infiltration of adipose tissue by immune cells and signals most epidermal growth factor receptor (EGFR) ligands that have been associated with hypertension, atherogenesis, vascular dysfunction, and cardiac remodeling [28,29].

Similarly, ADAMS 17 expression and TNFα levels were found to be increased in peripheral blood mononuclear cells, especially from individuals with advanced congestive heart failure [30]. Likewise, ADAMS 17 was associated with the development of atherosclerosis and the process of cardiac remodeling after AMI [16]. On the other hand, in Takotsubo Cardiomyopathy (TTC), hyperstimulation of beta-adrenergic receptors (β-ARs) resulting from the excessive release of catecholamines induced intracellular kinases capable of phosphorylating and activating ADAMS 17 [31].

### 3.1.10. MMP 1

The results concerning MMP 1 indicated a potential mechanistic link between this matrix metalloprotease and vascular remodeling in the context of endothelial dysfunction and thrombogenesis. MMP 1, a key enzyme involved in the degradation of type I collagen, may have played a pivotal role in destabilizing the ECM of atherosclerotic plaques, increasing its susceptibility to rupture, and therefore precipitating acute cardiovascular events. Although the study addressed the dysregulation of VWF and ADAMTS 13, MMP 1 is highlighted as a potential biomarker for atherosclerotic progression, particularly in relation to plaque vulnerability and ischemic mortality [19]. It is also involved in the degradation of the LDL receptor (LDLr), and its expression is regulated by TNFα; therefore its activity is enhanced during inflammatory processes such as atherosclerosis [32].

### 3.1.11. MMP 2

It has been demonstrated that interleukin-17 (IL-17) induced the expression of metalloprotease genes such as MMP 2 and MMP 9, which were implicated in the pathophysiology of atherosclerosis, suggesting a significant inflammatory role in the progression of this disease [33]. This molecule played a role in angiogenesis and adipogenesis. MMP 2, along with MMP 9, were produced by adipose tissue and were involved in modulating the ECM. This modulation is crucial for processes such as the differentiation of preadipocytes and the maturation of micro vessels, which are essential for adipogenesis and the expansion of adipose tissue [34].

### 3.1.12. MMP 8/MMP 9

The study results showed that RANK-L induced MMP release and was dependent on the activation of defined intracellular signaling pathways (PI3K/Akt and ERK1/2). In asymptomatic subjects, serum levels of RANK-L, MMP 8, and MMP 9 were positively correlated with coronary calcium scores, reflecting a relationship between circulating RANK-L and coronary calcification [35].

### 3.1.13. MMP 10

MMP 10, also known as stromelysin-2, was involved in vascular remodeling and the formation of unstable plaques, promoting inflammation and endothelial damage. An association was found with inflammation in asymptomatic subjects with cardiovascular risk factors, specifically atherosclerosis. A correlation was found between the atherosclerotic risk factors such as age, diabetes, and smoking with increased MMP 10 levels as the number of risk factors rose. Additionally, a link was found with inflammatory markers such as fibrinogen and high-sensitivity C-reactive protein (hs-CRP) [36].

### 3.1.14. MT4-MMP/MMP 17

MMP 17 (MT4-MMP) was initially described in the context of breast cancer, but plays a critical role in various physiological processes, including tissue remodeling, embryogenesis, organogenesis, tissue regeneration, angiogenesis, wound healing and inflammation. It contributed to the degradation of the ECM, facilitating cell migration and the repair of damaged tissues. In the context of disease, MMP 17 is implicated in pathological processes, such as tumor progression, metastasis, osteoarthritis and atherosclerosis, as well as in thoracic aortic aneurysms and dissections due to its ability to modulate cell behavior and the tumor microenvironment. These multifaceted roles underscore the importance of MMP 17 in both normal physiology and disease pathology [37].

### 3.1.15. Neprilysin

Neprilysin (NEP) is a promiscuous zinc metalloprotease. Low levels of soluble neprilysin may be linked to diastolic dysfunction and an adverse cardiometabolic profile. In another study neprilysin showed an involvement in proteolysis of numerous peptides, including natriuretic peptides. Paradoxically, it is of prognostic and therapeutic importance in heart failure with reduced ejection fraction, in which neprilysin was not correlated with natriuretic peptide levels and is not independently associated with adverse outcomes [38,39].

### 3.1.16. CD10

CD10 degrades a variety of vasoactive peptides, including vasodilators such as natriuretic peptides, bradykinin, and adrenomedullin, as well as vasoconstrictors like angiotensin II and endothelin-1 [40].

*3.2. Therapeutic Potential of ADAMTS and MMPs*

3.2.1. ADAMTS 2, 3 and 6

These proteins can be used as TGFB inhibitors accrediting them therapeutic potential benefits. They can also be used as circulating biomarkers of fibrosis in patients with CVD [10].

3.2.2. ADAMTS 7

It is an attractive drug target for vascular disease. Their pro-atherogenic and antiendothelial roles could be counteracted by systemic or local administration (such as through drug-eluting stents) of blocking agents ADAMTS 7 [11]. ADAMTS 7 overexpression can be effectively inhibited by TGFB, miRNA29a/b, and then effectively delay the development of cardiovascular disease [12].

3.2.3. ADAMS 10

Results suggest that calcitriol has therapeutic potential in RAGE-mediated cardiovascular treatment [14]. Studies have shown that inhibition of ADAMS 10 and ADAM17 decreases inflammation via peroxisome proliferator pathway [16].

3.2.4. ADAMTS 13

Modified forms of rADAMTS13, such as excised C-terminals dominions that can eliminate common autoantibody epitopes by affecting activity are being investigated [22]. ADAMTS 13 was proposed as a therapeutic agent to control TTP and treat other thrombotic complications through the direct thrombolytic effect on VWF-rich thrombi [21]. PLEX was only initiated in patients with a high probability of TTP and was discontinued when baseline ADAMTS 13 activity was >11% [23]. However, neither ADAMTS13 levels nor VWF/ADAMTS 13 ratio exhibited a significant impact on long-term adverse outcomes in ACS and spontaneous coronary artery dissection [20].

3.2.5. ADAMS 17

Rhom 1-2 are modulators of ADAMS 17 activity, responsible for regulating the elimination of ADAMS 17-dependent TNF in immune cells [28]. Studies have shown that inhibition of ADAMS 10 and ADAMS 17 decreases inflammation via peroxisome proliferator pathway [16].

3.2.6. MMP 1

As MMP 1 contributes to LDLr degradation, inhibiting it increases LDLr levels on the cell surface and enhances LDL uptake. This could be a potential strategy to lower LDL cholesterol and prevent cardiovascular disease [32].

3.2.7. MMP 2

Nanomaterials have become essential components in vaccine formulations and treatments for inflammation-driven CVD. They demonstrate therapeutic effects, notably through the reduced expression of MMP-2 in vitro in smooth muscle cells. This reduction suggests a potential mechanism by which nanomaterials could mitigate inflammatory responses associated with cardiovascular conditions [41].

3.2.8. MMP 8/MMP 9

ERK 1/2 and PI3K/Akt-dependent intracellular pathways could be promising therapeutic targets for reducing RANKL-mediated depression [35].

### 3.2.9. MMP 10

Although it has not been determined whether MMP 10 levels can predict the development of CVD, inhibiting MMP 10 may represent a promising therapeutic strategy to prevent the progression of atherosclerosis and improve cardiovascular outcomes, particularly in patients with risk factors such as diabetes and smoking. However, further research is needed to establish specific approaches for treating conditions related to MMP-10 [36].

### 3.2.10. MT4-MMP/MMP 17

Elevated MMP 17 levels may be associated with conditions like atherosclerosis, where its activity affects plaque stability and vascular integrity. These interactions highlight MMP 17 as a potential target for therapeutic interventions in CVD [37].

### 3.2.11. Neprilysin

According to the results of the analytical article by Yogesh et al., low levels of soluble neprilysin and an adverse cardiometabolic and smoking profile warrant further investigation [39]. Additionally, given the findings of degradation and inactivation by neprilysin resulting in vasoconstriction and elevated blood pressure levels, NEP has been targeted for its inhibition to elicit an antihypertensive response in patients with high blood pressure [42].

### 3.2.12. CD10

Shudong Wang et al. narrative review showed that inhibiting CD10 can enhance the effects of naturally natriuretic peptides, which help promote natriuresis, induce vasodilation, and decrease cardiac hypertrophy and fibrosis in heart failure patients. Also, soluble CD10 can degrade natriuretic peptides, offering a potential treatment strategy for heart failure. Moreover, the expression levels of membrane CD10 in neutrophils and granulocytes are correlated with heart failure prognosis, with lower expression linked to increased severity and higher expression associated with better outcomes [40].

## 4. Discussion

The involvement of MMPs in some CVD as biomarkers and therapeutic targets has led us to study the advances of these molecules in the literature. Regarding the date of publication, it was found that for the period from 2010 to 2023 the number of publications doubled, hence, it is possible that this increase is due to emerging molecular biology techniques that allow studying their diagnostic and therapeutic potential (Table S2).

In relation to ADAMTS 13, six documents associated decreased concentrations with negative effects such as acute myocardial infarction (secondary to increased platelet thrombi), reperfusion injury and infarction size [19–24]. Two studies reported the relationship between TTP and ADAMTS 13 deficiency, making it a biomarker for microangiopathic diagnosis, associated with an increased risk of even worse cardiovascular disease outcomes, including ischemic stroke and myocardial infarction [21,23]. Additionally, in six of the studies, an association between elevated levels of VWF with decreased ADAMTS 13 and high platelet reactivity with coronary artery disease was established, evidencing a mortality risk 1.46 times higher than people with high ADAMTS 13 activity [19].

In a study of patients with congenital heart disease, a decrease in ADAMTS 13 and VWF was identified because of the chronic consumption of these enzymes and the direct thrombolytic effect on VWF-rich thrombi [27]. However, the 41635A-allele of the ADAMTS 13 gene was associated with polymorphisms in the regulation of VWF, producing higher levels of this molecule, generating unwanted prothrombotic effects; and patients with the 1342G-allele had significantly higher frequency of atrial fibrillation and cerebral ischemic events. This happens both independently and in combination with VWF, which is consid-

ered a marker of endothelial activation, and its elevated levels have been associated with an increase in cardiovascular risk [26].

Regarding ADAMS 17, eight articles report its increase is related to endothelial damage. One of them involved in the pathophysiology of Takotsubo cardiomyopathy, evidencing acute myocardial inflammation and lipid dysregulation during acute stress triggered by a type I transmembrane protease known as ADAMS 17. This kinase-activated metalloproteinase through its intracellular phosphorylation, is widely expressed in the heart as membrane-bound proteins, so excessive increase in the ADAMS 17 cleavage process will trigger the cascade of acute cardiac inflammation, myocardial lipotoxicity, and low energy production [31]. In contrast, two articles demonstrated an association between TNFα and ADAM17 in patients with acute myocardial infarction (AMI) and atherosclerosis [30,31]. Additionally, only one article mentions the implication of a therapeutic potential through the inhibition of ADAMS 17 by iRhom2 (rhomboid pseudoprotease, involved in the degradation of epidermal growth factor), which has the function of regulating the inactivation and selective elimination of TNF in immune cells [28].

In one of the above-mentioned articles on ADAM 17 and ADAM 10 published in 2023, the dysregulation of ADAMS due to chronic kidney disease is recognized and shown that they may have detrimental and protective roles in atherosclerosis [29]. ADAM 17 was associated with a direct relationship between the development and resistance of atherosclerosis, which evidenced opposite effects on the formation of atherosclerosis between myeloid and endothelial ADAM 17. As in the study by Nicolaou et al. [43], which showed to have an atheroprotective role in the development of atherosclerosis, while myeloid-specific ADAM 17 deficiency resulted in an almost twofold increase in the size of atherosclerotic lesions with a more advanced lesion phenotype, characterized by a decrease in macrophages. Unlike other pathologies such as cancer, general inhibition of ADAM 17 results in unwanted side effects [13]. On the other hand, ADAM 10 is related to the development of atherosclerosis in humans; the expression of this disintegrant is significantly increased during atherosclerotic plaque progression compared to healthy human vessels and early atherosclerotic lesions [11].

It is worth highlighting the importance of a probable biomarker such as neprilysin, a proteolytic peptidase with beneficial and harmful effects that produces an inhibition of vasoconstriction, explaining the association with hypertension. On the other hand, a contrast was evidenced in two articles where one of them showed findings where the possible biomarker neprilysin does not demonstrate prognostic importance in heart failure with ejection fraction conservation (HFpEF), since in a cohort study of 144 patients it was found that neprilysin does not correlate with natriuretic peptide levels [42]. Additionally, an article showed that soluble neprilysin was not associated with death, heart failure, stroke, or myocardial infarction during a 10.7-year follow-up [39].

On the other hand, the articles showed divergence in terms of the pathophysiology of disintegrant and metalloproteinase ADAMS 10, some showing positive effects and others, negative and neutral effects. Firstly, its neutral effect is explained by the fact that the activation of this disintegrant and metalloprotease does not depend on the pathway and signaling of a molecule called iRhom, but depends on a molecular pathway involving tetraspanins, through the activation of Notch in several cell types. Unlike other molecules belonging to this family, such as disintegrant and metalloprotease ADAMS 17, which is involved in the pathogenesis of obesity by means of iRhom, generating a negative effect on the proliferation of adipocytes and an increase in cytokines that perpetuate the systemic inflammation characteristic of this pathology [28].

The positive effect of ADAMS 10 on CVD is evidenced by the modulatory action of calcitriol, reducing the expression of the advanced end products of glycosylation (RAGE)

receptor in cardiomyocytes, in the presence of the enzymatic activity of this disintegrant and metalloproteinase. Additionally, the initial negative impact of this molecule can be explained through its inhibition; this process generates an increase in ATII, which when binding to rAT1, mediates the transactivation of the epidemic growth factor receptor, releasing ligands that promote cell growth, specifically of cardiomyocytes, generating hyperplasia and hypertrophies of the cardiac chambers [14].

In four of the articles that mention the disintegrant and metalloproteinase ADAM 12, the literature showed a direct relationship between AngII and its receptor AT 1 with cardiovascular disease through vascular remodeling, causing hypertrophy, hyperplasia and migration of smooth muscle vascular cells. In this case, MMP releases epidermal growth factor ligands that, when binding with their receptor, generate transactivation, stimulating cardiomyocyte growth pathways and gene reorganization genes [15,18]. In this way, ADAMS 12 generates a negative effect through AT1 and G proteins, participating in the development of CVD such as hypertension, atherosclerosis and restenosis after vascular injury [44].

Additionally, it has been found that TGFB1 induces the production of ADAMS 12 through the expression of the gene with the same name. Likewise, the Son N molecule is a negative regulator of the TGFB1 signaling pathway and therefore of the expression of ADAMS 12, causing in cases of overexpression of Son N, the reduction in the production of ADAMS 12 through the decrease in TGFB1, which would lead to the stimulation of fibrosis and therefore the progression of certain pathologies such as cancer, CVD, central neural system (CNS) and musculoskeletal system diseases, suggesting a possible prognostic biomarker [17].

Regarding the three studies mentioning MMP 2, a negative effect was shown. First, it was evidenced that during the formation of an atheroma in the process of atherosclerosis, MMP 2 generated plaque rupture, causing fatal outcomes such as ischemic stroke and AMI [41]. Additionally, another result observed was that IL-17 induces the expression of metalloprotease genes such as MMP 2 and MMP 9, both molecules involved in the pathogenesis of atherosclerosis and platelet aggregation. The combination of IL-17 + TNF alpha resulted in a five-fold increase in invasive plate activity [33]. Secondly, the negative effect of MMP 2 in atrial fibrillation was seen, where in a cohort of patients treated with Thrombin, a decrease in this metalloproteinase was shown and activating cardiac fibroblasts to myofibroblasts, promoting arterial remodeling and consequently this cardiac arrhythmia [45].

It should also be noted that both MMP 2 and MMP 9 are cofactors associated with angiogenesis that are expressed in adipose tissue, thus one of the articles points out the importance of inhibiting adipokines, thus allowing to stabilize the production of MMP, managing to regulate angiogenic responses in vascular endothelial cells, understanding that there is a link between vascular growth and adipogenesis [34].

Continuing with MMPs 8 and 9, two articles observed that they have a negative effect on diseases such as atherosclerosis, obesity and diabetes mellitus. This effect could be explained by RANKL, which induces the production of MMP 8 and 9 by means of intracellular signaling cascades. Both molecules are related to coronary calcium, thus being possible biomarkers of the calcification process in the coronary arteries and in the underlying cardiovascular risk [43]. On the other hand, MMP 9, stimulated by some adipokines, such as Visfatin, is involved as a factor that produces angiogenesis in adipose tissue and is therefore considered to favor cardiovascular risk [34]. Continuing with MMP 10, its levels have been positively correlated with pro-inflammatory molecules such as fibrinogen, CRP and EIM (thickness of the intima media, in this case carotid); its elevated

levels perpetuate atherosclerotic plaques and some authors propose it as a useful biomarker to detect subclinical atherosclerosis in asymptomatic subjects [36].

Finally, referring to the therapeutic target as the objective of analysis in the articles proposed for this review, less than half of them analyze this variable. In this case, it is inferred that more studies are required in order to know its implications in the prevention and/or progression of CVD.

*Limitations*

Within the limitations of this study, certain factors must be considered when interpreting the results. Firstly, the majority of the included studies are narrative reviews, which lacked the methodological rigor of randomized controlled trials or meta-analyses. This limited the strength of the conclusions regarding causality or the therapeutic efficacy of metalloproteinase-targeted interventions in CVD. Furthermore, the heterogeneity in study designs, measurement techniques, and patient populations introduces challenges in synthesizing and directly comparing the results. Additionally, while studies published in English and Spanish were included, there is a potential selection bias due to the exclusion of studies in other languages, which could have provided further valuable insights.

## 5. Conclusions

Most of the studies in this review were classified as narrative reviews, being part of the first step of the evidence pyramid. This allowed us to demonstrate the clear need to include studies that provide more evidence on the subject, such as experimental studies, systematic reviews and meta-analyses. In cardiovascular pathology, there is diversity such as atherosclerosis, ischemic heart disease, thrombotic or thromboembolic disease, hypertension and cardiac hypertrophies. The relationship between MMPs and CVD was documented; the stimulation or inhibition of these molecules may lead to beneficial and/or adverse effects, which influence the progression and prognosis of the disease. Based on the literature available at the time, the vast majority were associated with diseases such as atherosclerosis, acute myocardial infarction and heart disease. More studies are needed to understand the pathophysiology and role of MMPs in CVD, as well as to focus efforts on research into biomarkers and future therapeutic targets.

**Supplementary Materials:** The following supporting information can be downloaded at: https://www.mdpi.com/article/10.3390/cardiogenetics15020014/s1. Table S1: Summary of findings; Table S2: Results according to the time period from 1996–2010 to 2011–2025.

**Author Contributions:** Conceptualization, C.K.P. and L.M.O.B.; methodology, L.M.O.B.; investigation, D.M.D.Q.; writing—original draft preparation, C.K.P. and L.M.O.B.; writing—review and editing, J.M.E.Á.; supervision, J.C.M.L. and A.M.S.G.; project administration, C.A.P.C. All authors have read and agreed to the published version of the manuscript.

**Funding:** There was none monetary funding from any institution and/or individual. The project was registered under code MED-346-2023 in Universidad de La Sabana, Chía Cundinamarca, Colombia. The University played a fundamental role in sponsoring the manuscript's development, providing institutional support to carry out the research. They collaborated with the research team by offering expert guidance, access to facilities, and enabling the use of databases and bibliographic resources.

**Data Availability Statement:** As this is a scoping review, no primary data were collected. The review is based on previously published studies, which are publicly available in peer-reviewed journals, databases, and repositories. All data used in this review are accessible through these sources. A list of the studies included in the review, along with relevant details, is provided in the Supplementary Materials or can be made available upon request.

**Acknowledgments:** We thank the University of La Sabana for its administrative and technical support.

**Conflicts of Interest:** The authors declare no conflicts of interest.

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
