# Peer review of "Pathophysiological Bases and Clinical Uses of Metalloproteases in Cardiovascular Disease: A Scoping Review"

_cardiogenetics, doi:10.3390/cardiogenetics15020014_

Round 1

Reviewer 1 Report

Comments and Suggestions for Authors

Dear Authors,

The text is well written, but if some tables were added it would be more usable, tables that underline the role of the various agents

By slightly modifying the search keywords, how many other bibliographic items would you find? only 50 seem few to me, comment

Reviewer 2 Report

Comments and Suggestions for Authors

The authors have conducted a comprehensive and well-structured review of the pathophysiological roles and clinical applications of metalloproteases in cardiovascular disease. The literature review is thorough, covering studies published up until 2023. My only suggestion is for the authors to review publications from 2024 and incorporate any relevant studies to ensure the most up-to-date insights are included.

Reviewer 3 Report

Comments and Suggestions for Authors

In this interesting review, authors aimed to describe the existing literature on the role of MMP in the pathophysiology of CVD and its usefulness in clinical practice for the diagnostic and therapeutic approach. The quality of the review and its significance is high enough to be published. Before that, authors might consider some minor suggestions:

-        The use of abbreviations should be revised.

-        Conclusions in Abstract section are not clearly expressed (the beginning is unclear).

-        Results (lines 103-107): Additional 12 articles were excluded, authors could explain the reason.

-        Figure 1 would better serve its purpose at the beginning of the Results section.

-        Tables S1 and S2 were not included in the reviewer’s materials.

-        Answers to research questions (lines 68-74) are not provided in Conclusions section.

Round 2

Reviewer 1 Report

Comments and Suggestions for Authors

paper looks good